# Digital Devices Use and Fine Motor Skills in Children between 3–6 Years

**DOI:** 10.3390/children10060960

**Published:** 2023-05-28

**Authors:** Francesca Felicia Operto, Andrea Viggiano, Antonio Perfetto, Gabriella Citro, Miriam Olivieri, Valeria de Simone, Alice Bonuccelli, Alessandro Orsini, Salvatore Aiello, Giangennaro Coppola, Grazia Maria Giovanna Pastorino

**Affiliations:** 1Child Neuropsychiatry Unit, Department of Medicine, Surgery and Dentistry, University of Salerno, 84081 Salerno, Italy; aviggiano@unisa.it (A.V.); antonio.perfetto92@gmail.com (A.P.); g.citro38@studenti.unisa.it (G.C.); mir.olivieri@gmail.com (M.O.); valeriades@hotmail.it (V.d.S.); salvatore91aiello@hotmail.com (S.A.); giangennaro.coppola@gmail.com (G.C.); graziapastorino@gmail.com (G.M.G.P.); 2Pediatric Neurology, Pediatric University Department, University of Pisa, 56126 Pisa, Italy; al.bonuccelli@gmail.com (A.B.); aorsini.md@gmail.com (A.O.)

**Keywords:** digital devices, digital tools, digital screen, children, fine motor skills

## Abstract

(1) Background: The principal aim of our research was to explore the relationship between digital devices use and fine motor skills in children aged three to six years and to explore the effect of some socio-demographic factors. (2) Methods: we enrolled 185 children aged between three to six years. The parents of all the participants fulfilled a questionnaire to explore the digital device use, and their children performed a standardized test to assess fine motor skills (APCM-2). We performed the Spearman correlation test to explore the relationship between different variables. (3) Results: the children spent an average of 3.08 ± 2.30 h/day on digital devices. We did not find a significant association between the time of use of digital devices and fine motor skills (*p* = 0.640; r = −0.036). The youngest children experienced digital tools earlier than older ones (*p* < 0.001; r = 0.424) and they were also the ones who used digital tools more time afterwards (*p* = 0.012; −0.202). The children who had working parents spent more time on digital devices (*p* = 0.028; r = 0.164/*p* = 0.037; r = 0.154) and used digital devices earlier (*p* = 0.023; r = 0.171). (4) Conclusions: This data suggest that it would be useful to monitor the use of digital tools, especially in the very first years of life. Future studies are needed to further explore this topic.

## 1. Introduction

In recent years, digital technologies have progressively become an integral part of the life of children and adolescents as indispensable tools for communication, relationships, and learning [1,2,3,4].

In particular, the “intuitive” technology of the new digital tools (e.g., smartphones and tablets) is very attractive and greatly stimulates the curiosity of younger children, that, with a single gesture of the finger, can access screens, images, videos, and sounds [5].

A recent Italian survey showed that 97% of children under 36 months and 89% under 18 months have already used digital device at least once [6]. Frequently, the parents use digital media as “pacifiers” (to keep calm the children, during meals, and before bedtime) [7,8,9,10,11,12,13]. 

The COVID-19 pandemic and its social consequences have widened digital device use and further lowered the age of those that access digital technologies, reinforcing a trend already underway in recent years [14,15,16,17,18]. 

Digital tools offer children many growth opportunities, but, at the same time, they can lead to risks. Some authors suggested that the use of digital tools constitutes a “passive” behavior, which takes the child away from other more useful learning experiences [19,20,21]. According to this, some recent reviews suggest that excessive and too early use of digital tools in children may be associated with delays in some cognitive, linguistic, and social skills [22,23,24]. 

On the other hand, some research suggests that the use of digital tools may have benefits in terms of learning and development for children, depending on many factors. It seems that the factors correlated to better learning are the use of interactive tools (e.g., touch-screen, tablet, etc.), rather than passive ones (e.g., television), as well as the use of so-called “educational” applications, co-viewing with parents, and a higher age in children. [25,26,27,28,29]. 

From the point of view of the scientific community, there are indications on generic exposure to digital technologies in the pediatric age, but not everyone agrees on the effects on psycho-physical wellbeing, which are not unique, depend on many factors, and can be both positive and negative.

The World Health Organization’s 24 h movement guidelines do not recommended screen time in children ≤1 year and suggested no more than 1 h of screen use in children between two to four years (less is better) [30].

The official recommendations of the Italian Society of Pediatrics on the use of media devices (mobile phones, smartphones, tablets, PCs, etc.) in children aged zero to eight take into consideration both the positive and negative effects on physical health and mental health of boys and girls: technology should not be rejected, on the contrary it should be used to the fullest, for example, by avoiding the use of smartphones to calm or distract the children [31]. The recommendations strongly advise the use of quality apps (e.g., educational games), but also indicate the times for exposure to digital devices in relation to the age and development of the children. It is not recommended to expose to smartphones and tablets before the age of two, during meals, and before going to sleep, namely, when the tablet or mobile phone should perform the function of calming or silencing the expression of needs. It is preferable to limit the use of these devices to a maximum of 1 h/day in children between two to five years or 2 h a day for those between the ages of five and eight [31].

The development of motor skills in children evolves gradually, according to well defined phases that begin in the first months of the child’s life [32]. In particular, fine motor skills require the coordination of small muscles of the hands, fingers, and eyes, and they allow us the movements necessary for some daily actions, such as handwriting, drawing, building small objects, typing, buttoning clothes, etc. [32]. The achievement of motor development milestones at the appropriate age allows the clinician to monitor the correct development of the child. Some studies suggest an association between fine motor coordination skills in children and later academic performance, and a delay in acquiring fine motor developmental milestones may be indicative of cognitive or medical problems and should be further explored [33,34,35].

The relationship between fine motor skill development and digital device use in children is currently under debate. However, the evidence of recent the literature remains inconclusive, and there is a lack of studies specifically focused on the different content of digital devices.

The recent review by Arabiat and colleagues (2022) [36] reported that interactive digital technology was negatively associated or unrelated to motor skills. Moon et al. (2019) [37] reported that smart device use was positively related with fine motor skills in children aged three, but no significant relationship was found for children aged four to five years.

Other evidence of observational studies suggests a negative association between digital tools use in children and fine motor abilities [38,39,40,41]. On the contrary, other evidence showed an improvement in fine motor skills in children who use new digital tools (e.g., tablets) [42,43,44,45]. Probably, these discordant data are the consequence of the heterogeneity of the studies and of the difficulty to accurately measure the exposure time to digital devices. However, most studies agree that “active” digital technologies (tablets, smartphones, etc.) have a better impact on children’s developmental skills than “passive” ones (television, video viewing, etc.). Furthermore, high-quality learning apps would be preferable, and co-viewing by an adult is very important to mediate the use of the device in order to improve the child’s learning experience.

Based on this evidence, the principal aim of our research was to evaluate the relationship between fine motor skills and digital device use in children aged three to five years. The secondary aim was to explore the modality of use of digital tool in this age group and to evaluate how some socio-demographic variables can affect the relation between fine motor abilities and digital device use. 

## 2. Materials and Methods

### 2.1. Participants

Our cross-sectional observational study explored digital devices use and fine motor abilities in children, recruited from four kindergartens of the city of Salerno (Italy).

We included all the children aged between 37–72 months; the exclusion criteria were: (i) medical or neuropsychiatric diagnoses that could affect neuropsychomotor development (autism spectrum disorder, intellectual disability, language disorder, psychomotor development delay), (ii) poor child compliance to perform neuropsychological evaluation, and (iii) poor parental compliance.

The parents of all the participants had a preliminary meeting with a child neuropsychiatrist, who informed them about the methods and the purpose of the study. Written informed consent was collected for all the participants.

All the parents that took part in the study completed the Digital Devices Questionnaire, a non-standardized tool that investigated the frequency and the modality of digital device use by their children.

Subsequently, the children underwent a standardized neuropsychological evaluation, administered by two child neuropsychologists, aimed to assess fine motor skills (APCM-2–Abilità Prassiche e della Coordinazione Motoria–2a Edizione) [46].

Our study was conducted in agreement with the Helsinki declaration and was approved by the Campania Sud Ethics Committee (number of the protocol = 0033986 of 5 March 2019).

### 2.2. Questionnaire on Digital Device Use

The questionnaire on digital devices is composed of two sections: a first part, in which general data, socio-demographic information, and medical history (age, sex, household members, siblings number, parents’ educational background and employment, pregnancy, childbirth, psycho-motor progression, and medical conditions) were collected, and a second part examining the use of digital devices in children, based on 12 items, as follows:digital devices provided at homechildren’s favorite toolsage from when digital device use startedmedium daily use timemodality of use (with/without parents’ monitoring)chosen content (with/without dialogues)parent control (independent choice of children/choice guided by the parent)behaviors implemented through the use of digital tool (level of frustration, reaction to names, social attention)reasons for that the parents allow digital devices to their child (to entertain, to calm, during sleep time/meal time)parents’ perception of the risks associated with digital device use for their childrenparents’ seeking advice from pediatrician regarding use of digital device in childrentime that the child spends in social activities with their peers.

### 2.3. APCM-2—Abilità Prassiche e Della Coordinazione Motoria—2a Edizione

The test “Abilità Prassiche e della Coordinazione Motoria (Praxic and Motor Coordination Skills)—2nd Edition” (APCM-2) is an Italian test developed and validated by Sabbadini [46]. This test aims to assess motor and praxis skills in children aged between 2–8 years. This tool includes 6 forms, divided into different by age subgroups: 24–36 months (short version and complete versions), 37–48 months, 49–60 months, 61–72 months, and 6.1–8 years. This test includes the evaluation of two scales: (a) motor strategies (eye motions, movements of hand and fingers, balance and coordination, sequencing), and (b) adaptive cognitive processes (symbolic gestures, constructive praxis abilities, manual skills, graphomotor skills, dynamic coordination). In our research, we considered only the last scale mentioned. Raw scores were converted into weighted scores and percentiles scores (a score <5° percentile was considered under the norm).

The standardization sample comprised 261 children aged from three–eight (54% = boys, 46% = girls). Internal consistency for both scales (motor strategies and adaptive cognitive processes) was higher than 0.75.

### 2.4. Statistical Analysis

We expressed the continuous variables as mean ± standard deviation and categorical variables as proportions/percentage.

We performed the Shapiro-Wilk normality test to verify the data distribution. Due to the presence of data with non-normal distribution, we employed non-parametric tests for our analysis. We performed a Spearman correlation test and multivariate regression analysis in order to explore the relationship between the different variables; *p* < 0.05 was considered statistically significant.

Statistical Package for Social Science, version 23.0 (IBM Corp, 2015) was employed to analyze all data.

## 3. Results

### 3.1. Sample Characteristics

We recruited 185 children (105 males = 57%; 80 females = 43%) aged between 37–72 months (mean = 55.02 ± 9.50). All kindergarten children were recruited (except two children with certified disabilities). All the subjects recruited agreed to participate in the study. All parents filled in the questionnaires, and all children had good compliance with the APCM-2 test, so there was no missing data.

The socio-demographic features of the sample are reported in Table 1.

### 3.2. Use of Digital Devices by Children

In our sample, all the children (185/185, 100%) used of at least one digital device, as reported by their parents, and the digital devices that they preferred were television (134/185, 72%), smartphone (132/185, 71%), and tablet (75/185, 41%). 

The first digital tool that children used was television (mean age = 17.15 ± 8.99 months), followed by smartphone (29.56 ± 13.30 months) and tablet (31.66 ± 12.41 months).

In our sample, the children spent a total mean time on digital device of 3.08 ± 2.30 h/day, preferring television (1.57 ± 1.22 h/day) and smartphone (0.88 ± 1.02 h/day). 

In our sample, the parents reported that most of the children used the digital devices with their parents (119/185, 64%) or brothers/sisters (21/185, 11%) or with parental control (31/185, 18%), and an amount of 11% of children used digital devices alone (21/185). 

All the children preferred contents with dialogues (185/185, 100%), and almost half of them played videogames (91/185, 49%). 

The parents allowed their children to use digital devices for entertainment (129/185, 70%), during mealtimes (56/185, 30%), to calm the child (46/185, 25%), and before the child went to sleep (24/185, 13%). 

Some parents reported absorbent behaviors in their child when they used digital devices. In particular, some children did not respond when called (10/185, 5%), did not interact with others (3/185, 2%), or appeared frustrated when the parents tried to lure them away from the device (14/185, 7%).

Sleep problems are reported in about 19% (36/185) of the children.

Finally, 43% of parents (80/185) were concerned about the consequences of the digital device use on their children health, but only 18% of them (34/185) turned to the pediatrician for advice. These results were summarized in Table 2.

### 3.3. Relation between Digital Devices Use and Fine Motor Skills in Children

We did not find statistically significant relationship between the APCM-2 scores and digital device daily time use (smartphone, tablet, personal computer, videogames) (Table 3). 

We found a positive significant relationship between start age of use smartphone/tablet and the APCM-2 weighted score (Table 3; Figure 1).

To exclude the effect of age on this relationship, we performed a multivariate regression analysis. Adding age as a covariate the relationship between start age of use and the APCM-2 weighted score is no longer statistically significant (smartphone *p* = 0.109, ß = 0.239; tablet *p* = 0.056, ß = 0.393; total *p* = 0.297, ß =0.178).

We found that total daily time of digital device use, daily time of smartphone use, and daily time of television use were negatively related to age to start using digital device (Table 4; Figure 2). 

Daily time of smartphone use was positively related to the mothers’ and fathers’ jobs and negatively related to the fathers’ degree (Table 4).

Daily time of personal computer use was negatively related to the mothers’ degree (Table 4).

Daily total time of all digital device use was positively related to the mothers’ and fathers’ jobs (Table 4).

The age of start to use all digital device was positively related to the age of child, except for television (Table 5; Figure 3).

The age of starting use of tablets was positive related to the fathers’ and mothers’ degrees (Table 5).

The age of starting use any digital device was positive related to the fathers’ job (Table 5).

## 4. Discussion

The principal aim of our research was to explore the digital devices use in children aged between three to five years and their relationship with fine motor abilities. We considered a sample of 185 typically developing children (57% male; mean age = 55.02 ± 9.50 months; Table 1) that performed a standardized test that assessed fine motor abilities (APCM-2), while the modality of use of digital device was explored through a questionnaire addressed to their parents.

Regarding the first question of our study, we did not find a significant relation between time of digital device use and fine motor abilities in our sample (Table 3). In particular, the time spent on digital devices was not significantly associated with neither an improvement, nor a deterioration, of performance in grapho-motor skills and manual coordination. We must underline that we have not performed qualitative analyses on the modality of digital device use (use of “educational” applications, “active” or “passive” use of digital tools, and possible parent/caregiver involvement during the use). 

We found a relationship between the first age of use of digital device and fine motor skills (Table 4) (the children who used digital tools early had worse fine motor performance). However, this result cannot be considered significant because it is affected by the effect of the age of children on this relationship.

We also found that some socio-demographic characteristics were associated with the daily time of use of digital tool, such as the age of first use (Table 4). Interestingly, the correlation analysis showed that children who used digital devices earlier were those who spent more hour/day with them. It would be interesting, in future studies, to explore the relationship between the first age of use of digital tools by children and the time spent with them in the school age and in adolescence, in consideration of the growing phenomenon of “digital addiction”.

In addition, we found that the start age of use of digital device was also related to the age of the child (Table 5), confirming the phenomenon of “digital anticipation”, because younger children use digital technologies more and more precociously [1,2,3,4].

Another significant correlation was found between the daily time on digital devices/age of start to use digital device and the parent’s job. The children who had working parents spent more time on digital devices (Table 4) and used digital devices earlier (Table 5). In this regard, we must not forget the fundamental role played by parents in digital device exposure [12,13].

Finally, the correlation analysis did not show significant differences between males and females in the modality and in times of use of digital devices, except for video games, which were preferred by males (Table 4 and Table 5). 

Our research agreed with the cross-sectional study of Moon and colleagues (2019) [37]. The authors enrolled 117 children between three to five years, divided into three age groups, and administered to their parents a questionnaire to evaluated the relationship between smart device use and language, as well as motor and social skills. In children aged three years, the authors found that smart device use was positively related with fine motor skills and negatively associated with language, but no significant relationship was found for children aged four to five years.

Contrary to our study, there is some evidence of both negative and positive correlation between digital device use and fine motor abilities.

The review of Arabiat and colleagues [37], which considered 53 studies on children under seven years, concluded that children’s use of interactive digital tools was positively associated with language abilities and negatively associated or unrelated to motor skills. Furthermore, the authors highlighted that the measurements of intensity and duration of digital device exposure were often imprecise. 

The study of Zheng and Sun (2021) [41], which involved 877 children aged three-six years, concluded that the time spent on digital tools negatively predict fine and gross motor skills of children (particularly the passive use was negatively associated to school performance and fine motor abilities).

Kiefer et al. (2015) [38] considered 23 preschool children divided into two groups, the first group of 11 children using the keyboard and the second of 12 children using handwriting, with the aim of determining whether there was a difference between the two groups in writing skills. The authors found that the handwriting group had better writing skills.

Lin et al., (2017) [39] monitored 40 preschool children who used tablets for more than 60 min per week for 24 weeks, comparing them to 40 age- and gender-matched children participating in a hand skills program. The authors found that motor performance improved only in the control group.

In another study by Lin (2019) [40], the authors compared 36 tablet-using children and 36 non-tablet-using preschoolers, showing that the children in the non-tablet group performed significantly better in fine motor tasks.

Some studies also found some benefits in fine motor abilities of children using an iPad [42] or notebook application [44]. On the contrary, Axford et al. (2018) [42] suggested no benefits for children that use an iPad over 30 min/day.

The cross-section study of Souto et al. (2020) [45] considered 78 children aged between 24–48 months, divided into two groups: frequent use of tablet (n = 26) and no experience of tablet use (n = 52). The Bayley Scale of Infant Development III was performed to explore the relationship between fine motor abilities and interactive tablet use. Most of the children performed both active and passive activities, usually accompanied by their parents, not exceeding the recommended times. The major finding of the study was that the children using tablet performed slightly better than the other group.

This apparent discrepancy between our results and those found in the literature may be due to the fact that, in our study, we only considered the total time of digital device use, but we did not consider other factors, such as the employing of “educational” applications and the role of parents while children used the digital tools.

Analyzing the digital device questionnaire, we found that all the children used at least one of the digital tools (smartphones, tablets, television, personal computers, or videogames) (Table 2). These data agree with the recent literature studies, according to which the use of digital tools in pre-school age has greatly increased, reflecting both the increasing use of digital tools by families and contemporary society [1,4].

The children preferred television (72%), smartphone (71%), and tablet (41%) (Table 2). Comparing these data with our previous research, we can observe an increase in the preference for the Tablet in children aged three to five years vs. children under three years [6]. 

The digital tool used first by children was television (mean age of first use = 17.15 ± 8.99 months), followed by smartphone (29.56 ± 13.30 months), tablet (31.66 ± 12.41 months), PC (34.90 ± 16.74 months), and videogames (43.71 ± 16.88 months) (Table 2). These results, compared to those of children under the age of three, show that, in younger children, there is an increasingly precocious exposure to digital devices (for example, in children between the ages of three to five, the first use of the smartphone was at 29.56 ± 13.30 months, while, in children under three, it was 15.6 ± 5.8 months). This “anticipation” phenomenon had already been underlined in several previous studies [6,47,48].

The parents reported that the mean daily use of digital device was about 3 h/day, according to many recent survey [1,2,3,4] (Table 2). The most commonly used digital tools are television (1.57 ± 1.22 h/day), the smartphone (0.88 ± 1.02 h/day), and the tablet (0.43 ± 0.71) (Table 2). It is important to underline that, compared to children under the age of three, children in our sample spent an average of one hour more a day using digital tools [6], exceeding the limits recommended by scientific societies [30,31]. 

An amount of 64% of children used digital devices with parent/caregiver or siblings (11%), but 11% and 18% of them used digital tools alone or with automatic parental control, respectively (Table 2). 

The parents allowed their children to use digital devices to entertain (70%), during mealtime (30%), to calm down (25%), and to let the children sleep (13%), suggesting that the parents mainly using these tools as “peacekeeper” while they are engaged in other activities [12,13,49] (Table 2). 

Some parents reported atypical behaviors in their children during the digital device use: the children did not respond when called (5%), they did not interact (2%), and they appeared frustrated when the digital tool was take away (7%) (Table 2). This result confirms that digital devices use in some children can compromise social interaction and lead to an emotional deregulation [36,37].

In our sample, sleep problems were reported in 19% of children (Table 2). A relationship between sleep problems and excessive digital media use by children had already been highlighted [50,51]. 

An amount of 43% of parents say they are concerned about the digital devices use on their child’s health. However, only 18% of them asked pediatricians for advice on this topic (Table 2).

This study has several limitations. First of all, the use of parental questionnaires to monitor the modality of use of digital tools is not free from possible bias. Furthermore, the study only included children aged three to five years, which limits the generalizability of its findings to other age groups. Additionally, the study did not investigate all potential confounding factors that could influence the relationship between digital device use and fine motor skills. Finally, we did not consider other environmental factors that could impact childrens’ digital habits, such as time spent on digital devices by their parents, as well as the use of “educational apps”, and we did not differentiate between “active” or “passive” co-viewing. We, therefore, propose to explore these aspects in future research. 

The strength of our work is the sample size and the use of standardized and direct neuropsychological tests to assess fine motor skills in children. 

The results of our study have several practical implications, as they help to better define the possible impact of digital devices on the psychomotor development of children. Based on our results, it would be useful to monitor the time spent by children on digital tools, the age of first use, but also the modality of digital device use in order to ensure the most correct use possible. Our results highlighted, also, that most parents are concerned about the effect of digital tools on their children’s health, but only few of them ask for the support of pediatricians. Therefore, it would be important to implement the communication on this topic through information campaigns. In the future, it would be interesting to perform prospective studies that allow us to monitor the relationship between the digital devices use and other development skills (academic skills, as well as emotional and social-relational skills) during school age and adolescence, also with the aim of prevent the phenomena of digital addiction.

## 5. Conclusions

In our study, the children spent an average of about three hours a day on digital devices. We did not find a significant relationship between the time of use of digital devices and fine motor skills.

The youngest children experienced digital tools at an earlier age, and they were also the ones who used digital tools more time afterwards. This trend should be monitored over time in order to prevent the phenomenon of “digital addiction”, especially in adolescents.

Future studies are needed to confirm these data and further explore the relationship between digital device use and fine motor skills, taking into account other aspects, such as content of digital device. 

In the meantime, parents and pediatricians should be aware that the use of digital devices in children can bring both risks and benefits. Therefore, it is important not to exceed the time of use, to prefer active co-viewing with parents, and to encourage the use of quality educational applications.

## Figures and Tables

**Figure 1 children-10-00960-f001:**
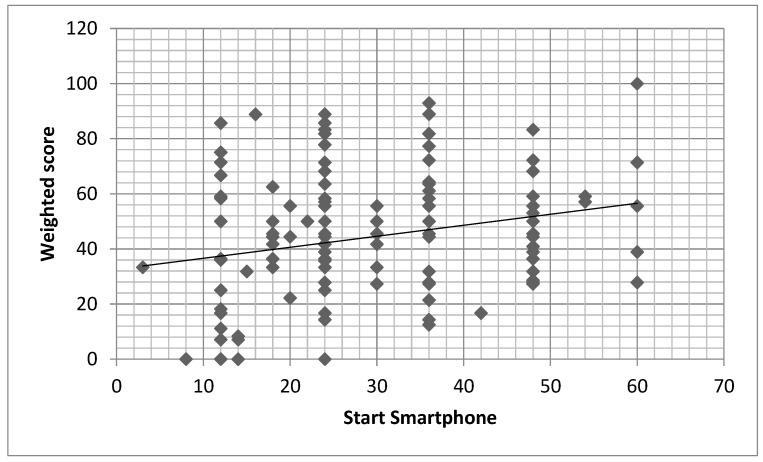
Relationship between age of start using smartphone and APCM-2 weighted score (r = 0.200; *p* = 0.009.). *x* axis = age of start using smartphone (in months); *y* axis = APCM-2 weighted score.

**Figure 2 children-10-00960-f002:**
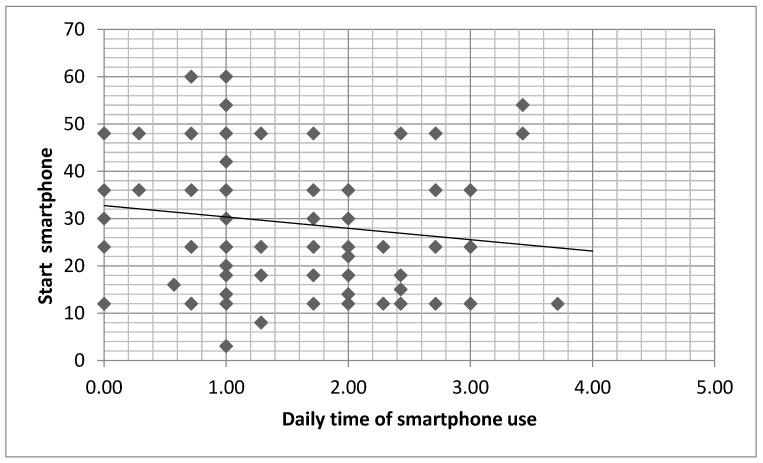
Relationship between age of start using smartphone and daily time of smartphone use r = 0.202; *p* = 0.012). *x* axis = Daily time of smartphone use (in hours); *y* axis = age of start using smartphone (in months).

**Figure 3 children-10-00960-f003:**
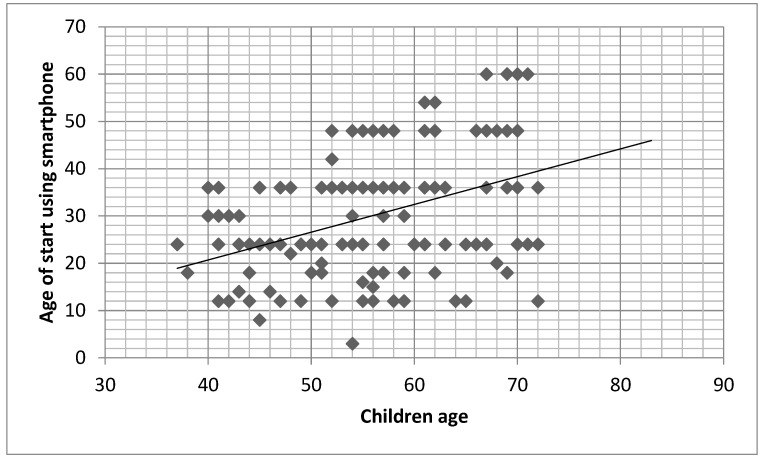
Relationship between children age and age of start using smartphone (r = 0.424; *p* < 0.001). *x* axis = children age (in months); *y* axis = age of start using smartphone (in months).

**Table 1 children-10-00960-t001:** Sample characteristics. m = mean; SD = standard deviation; MS = middle school; HS = High School; UN = University; U/H = unemployed/housewife; SW = skilled worker; OW/T = office worker, teacher; SE = self-employed.

	Total Sample			
**Sample Size**	n = 185			
**Sex**				
male	105 (57%)			
female	80 (43%)			
**Age—m ± SD**	55.02 ± 9.50 months			
**Pregnancy problems**	29 (16%)			
**Birth problems**	33 (18%)			
**Perinatal problems**	15 (8%)			
**Birth week—m ± SD**	38.67 ± 2.12			
**Birth weight—m ± SD**	3.16 ± 0.55 kg			
**Crawling**	44 (24%)			
**First step age m ± SD**	13.22 ± 2.5 months			
**First word age m ± SD**	14.85 ± 6.68 months			
**Medical pathologies**	29 (16%)			
**Family members**	4.08 ± 0.98			
**Siblings —m ± SD**	0.93 ± 0.85			
**Mother age—m ± SD**	37.64 ± 4.84 years			
**Father age—m ± SD**	41.23 ± 5.74 years			
**Educational level**	**MS**	**HS**		**UN**
Mother	21 (11%)	70 (38%)		94 (51%)
Father	32 (17%)	79 (43%)		71 (38%)
**Job**	**U/H**	**SW**	**OW/T**	**SE**
Mother	45 (24%)	20 (11%)	71 (38%)	47 (25%)
Father	1 (0.5%)	45 (24%)	69 (37%)	65 (35%)

**Table 2 children-10-00960-t002:** Questionnaire on digital device use; m = mean; SD = standard deviation; DD = digital devices; * (only for children ≥12 months, total sample size = 236).

	Total Sample		
**Sample Size**	**185**		
**Use of DD by children**			
**(at least one)**	185 (100%)		
**DD available at home**			
Smartphone	131 (71%)		
Tablet	100 (54%)		
Personal Computer	45 (24%)		
Television	175 (95%)		
Videogames/console	18 (10%)		
**Children’s favorite DD**			
Smartphone	132 (71%)		
Tablet	75 (41%)		
Personal Computer	17 (9%)		
Television	134 (72%)		
Videogames/console	17 (9%)		
**Age of start using DD**	months (m ± SD)		
Smartphone	29.56 ± 13.30		
Tablet	31.66 ± 12.41		
Personal Computer	34.90 ± 16.74		
Television	17.15 ± 8.99		
Videogames/console	43.71 ± 16.88		
**Time of use DD**	h/day (m ± SD)		
Smartphone	0.88 ± 1.02		
Tablet	0.43 ± 0.71		
Personal Computer	0.14 ± 0.50		
Videogames	0.06 ± 0.22		
Television	1.57 ± 1.22		
Total Time	3.08 ± 2.30		
**Modality of use**			
With parent/caregiver	119 (64%)		
With brother/sister	21 (11%)		
With parent-control	34 (18%)		
alone	21 (11%)		
**Favorite contents**			
With dialogue	185 (100%)		
Without dialogue	17 (6%)		
Videogames	91 (49%)		
**Contents selection**			
Independent choice	150 (81%)		
Choice guided by parents	35 (19%)		
**Reasons for granting DD**			
To entertain	129 (70%)		
To calm down	46 (25%)		
To let the child sleep	24 (13%)		
During mealtime	56 (30%)		
**Concerns about DD use**			
Yes	80 (43%)		
No	105 (57%)		
**Request to the pediatrician**			
Yes	34 (18%)		
No	151 (82%)		
**Behavior during DD use**			
	absent	partial	immediate
**response to name**	10 (5%)	110 (59%)	67 (36%)
	absent	partial	adequate
Social attention	3 (2%)	31 (17%)	150 (81%)
	high	medium	low
Frustration	14 (7%)	83 (45%)	90 (49%)
**Social activities** *			
Less than once a week	9 (5%)		
About once a week	48 (26%)		
Several times a week	91 (49%)		
Often or every day	41 (22%)		
**Sleep problems**	36 (19%)		

**Table 3 children-10-00960-t003:** Spearman correlation test between fine motor skills and use of digital device. *p* < 0.05 are in bold.

	Daily Time of Digital Device Use					
	Smartphone	Tablet	Computer	Television	Videogames	Total
**APCM-2 (fine motor skills)**						
Weighted score	r = −0.095 *p* = 0.213	r = 0.014 *p* = 0.851	r = 0.007 *p* = 0.931	r = 0.007 *p* = 0.922	r = −0.029 *p* = 0.703	r = −0.036 *p* = 0.640
Percentile score	r = −0.075 *p* = 0.326	r = 0.005 *p* = 0.952	r = 0.074 *p* = 0.330	r = 0.026 *p* = 0.731	r = 0.004 *p* = 0.958	r = −0.002 *p* = 0.983
	**Start Age of Digital Device Use**					
	**Smartphone**	**Tablet**	**Computer**	**Television**	**Videogames**	**Total**
**APCM-2 (fine motor skills)**						
Weighted score	r = 0.219 *p* = 0.008	r = 0.277 *p* = 0.008	r = 0.313 *p* = 0.098	r = 0.157 *p* = 0.064	r = 0.354 *p* = 0.235	r = 0.200 *p* = 0.009
Percentile score	r = 0.046 *p* = 0.584	r = 0.129 *p* = 0.226	r = 0.221 *p* = 0.249	r = 0.091 *p* = 0.284	r = −0.120 *p* = 0.695	r = 0.044 *p* = 0.564

**Table 4 children-10-00960-t004:** Spearman correlation test between daily time of digital device use and other socio-demographic characteristics. *p* value < 0.05 are in bold.

	Daily Time of Use DD					
	Smartphone	Tablet	Computer	Television	Videogames	Total
**Start Age digital Device Use**	r = −0.321 *p* < 0.001	r = 0.074 *p* = 0.480	r = −0.060 *p* = 0.755	r = −0.183 *p* = 0.023	r = −0.270 *p* = 0.350	r = −0.202 *p* = 0.012
**sex**	r = −0.144 *p* = 0.051	r = 0.102 *p* = 0.168	r = 0.068 *p* = 0.361	r = −0.095 *p* = 0.198	r = 0.155 *p* = 0.036	r = 0.056 *p* = 0.445
**age**	r = 0.040 *p* = 0.587	r = 0.068 *p* = 0.357	r = −0.069 *p* = 0.354	r = −0.096 *p* = 0.194	r = 0.111 *p* = 0.132	r = 0.034 *p* = 0.641
**Father age**	r = −0.038 *p* = 0.609	r = 0.012 *p* = 0.868	r = −0.005 *p* = 0.944	r = 0.668 *p* = 0.362	r = 0.046 *p* = 0.537	r = 0.021 *p* = 0.774
**Father Degree**	r = −0.157 *p* = 0.034	r = 0.028 *p* = 0.710	r = −0.132 *p* = 0.076	r = −0.011 *p* = 0.884	r = −0.036 *p* = 0.630	r = 0.068 *p* = 0.365
**Father job**	r = 0.205 *p* = 0.006	r = 0.100 *p* = 0.183	r = 0.058 *p* = 0.442	r = 0.030 *p* = 0.685	r = −0.015 *p* = 0.838	r = 0.164 *p* = 0.028
**Mother age**	r = −0.086 *p* = 0.251	r = −0.054 *p* = 0.468	r = 0.079 *p* = 0.292	r = −0.032 *p* = 0.666	r = 0.057 *p* = 0.446	r = −0.071 *p* = 0.342
**Mother degree**	r = 0.104 *p* = 0.157	r = 0.004 *p* = 0.955	r = −0.170 *p* = 0.021	r = −0.050 *p* = 0.500	r = 0.069 *p* = 0.353	r = 0.032 *p* = 0.670
**Mother job**	r = 0.218 *p* = 0.003	r = 0.073 *p* = 0.324	r = −0.053 *p* = 0.475	r = 0.003 *p* = 0.972	r = 0.048 *p* = 0.523	r = 0.154 *p* = 0.037

**Table 5 children-10-00960-t005:** Spearman correlation test between age of start to use digital device and other socio-demographic characteristics. *p* value < 0.05 are in bold.

	Start Age of Use DD					
	Smartphone	Tablet	Computer	Television	Videogames	Total
**Daily Time of Digital Device Use**	r = −0.321 *p* < 0.001	r = 0.074 *p* = 0.480	r = −0.060 *p* = 0.755	r = −0.183 *p* = 0.023	r = −0.270 *p* = 0.350	r = −0.202 *p* = 0.012
**sex**	r = −0.012 *p* = 0.886	r = 0.023 *p* = 0.824	r = 0.079 *p* = 0.678	r = 0.030 *p* = 0.721	r = 0.226 *p* = 0.437	r = 0.037 *p* = 0.621
**age**	r = 0.363 *p* < 0.001	r = 0.350 *p* = 0.001	r = 0.559 *p* = 0.001	r = 0.139 *p* = 0.092	r = 0.702 *p* = 0.005	r = 0.424 *p* < 0.001
**Father age**	r = 0.071 *p* = 0.380	r = −0.037 *p* = 0.724	r = 0.181 *p* = 0.348	r = −0.013 *p* = 0.876	r = 0.373 *p* = 0.189	r = −0.013 *p* = 0.859
**Father Degree**	r = 0.031 *p* = 0.705	r = 0.309 *p* = 0.003	r = 0.155 *p* = 0.423	r = −0.052 *p* = 0.536	r = 0.185 *p* = 0.526	r = −0.013 *p* = 0.859
**Father job**	r = 0.074 *p* = 0.367	r = 0.173 *p* = 0.097	r = −0.037 *p* = 0.856	r = −0.006 *p* = 0.947	r = 0.086 *p* = 0.779	r = 0.171 *p* = 0.023
**Mother age**	r = 0.123 *p* = 0.130	r = −0.070 *p* = 0.505	r = 0.183 *p* = 0.334	r = 0.145 *p* = 0.079	r = 0.010 *p* = 0.972	r = −0.007 *p* = 0.928
**Mother degree**	r = −0.046 *p* = 0.567	r = 0.271 *p* = 0.008	r = −0.239 *p* = 0.204	r = 0.126 *p* = 0.127	r = 0.167 *p* = 0.568	r = 0.082 *p* = 0.268
**Mother job**	r = 0.084 *p* = 0.303	r = 0.121 *p* = 0.244	r = −0.238 *p* = 0.204	r = −0.094 *p* = 0.256	r = 0.065 *p* = 0.832	r = 0.137 *p* = 0.066

## Data Availability

The data presented in this study are available upon request from the corresponding author.

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
