# Peer review of "Digital Devices Use and Fine Motor Skills in Children between 3–6 Years"

_children, 2023, doi:10.3390/children10060960_

Round 1
Reviewer 1 Report
The Authors of the manuscript conducted an interesting study on the impact of digital device use on fine motor skills. Given the fact that digital addiction is a growing problem, this research is extremely valuable. However, I have a few comments, below is the list:
Line 22 - please clarify the term 'an earlier age'
Lines 50-53 - please elaborate with examples for what age there will be a visible positive impact and whether any more factors need to be met. The research is too laconically described.
Lines 159-163 - the description of APCM-2 is very chaotic and incomprehensible
Line 208 - did the children appear frustrated at the time of DD use, or when the parents tried to get them away from the device or interested in something else? This is not accurately described
The question about time spent with DD by parents is missing from the questionnaire. As an influence of the environment on children's activities
Figure 1 and 2 - r should appear in the figures
There should also be a comparison of the survey items by age group. If these were done, they should also be in tabular form.
The discussion should include a summary table as there are a lot of results presented and a table included will bring some clarity to the data presented.
Author Response
Reviewer #1
Reviewer comment:
The Authors of the manuscript conducted an interesting study on the impact of digital device use on fine motor skills. Given the fact that digital addiction is a growing problem, this research is extremely valuable. However, I have a few comments, below is the list:
- Line 22 - please clarify the term 'an earlier age'
- Lines 50-53 - please elaborate with examples for what age there will be a visible positive impact and whether any more factors need to be met. The research is too laconically described.
- Lines 159-163 - the description of APCM-2 is very chaotic and incomprehensible
- Line 208 - did the children appear frustrated at the time of DD use, or when the parents tried to get them away from the device or interested in something else? This is not accurately described
- The question about time spent with DD by parents is missing from the questionnaire. As an influence of the environment on children's activities
- Figure 1 and 2 - r should appear in the figures
- There should also be a comparison of the survey items by age group. If these were done, they should also be in tabular form.
- The discussion should include a summary table as there are a lot of results presented and a table included will bring some clarity to the data presented.
Author response: we thank the reviewer for the comment and suggestion. We implemented and modified the manuscript as required:
- Line 22 – we changed the sentence into “The youngest children experienced digital tools earlier than older ones”
- Line 50-53 – we better specified the positive factor that impact on learning, as follow: “On the other hand, some research suggest that the use of digital devices may have benefits in terms of learning and development for children, depending on many factors. It seems that the factors correlated to better learning are the use of interactive tools (e.g. touch-screen, tablet etc) rather than passive ones (e.g. television), the use of so-called "educational" applications, co-vieving with parents and a higher age in children.”
- Lines 159-163 – we modified the description of APCM-2, as follow: The test “Abilità Prassiche e della Coordinazione Motoria (Praxic and Motor Coor-dination Skills)—2nd Edition” (APCM-2) is an Italian test created and validated by Sab-badini [46]. This test aims to assess motor and praxis skills in children aged between 2–8 years. This tool includes 6 forms of the questionnaire, divided by age groups: 24–36 months (short version and complete versions), 37–48 months, 49–60 months, 61–72 months, and 6.1–8 years. The test includes the evaluation of two scales: (a) motor schemes (balance and coordination, sequencing, ocular movements, hand and fingers movement), and (b) adaptive cognitive functions (dynamic coordination, symbolic gestures, con-structive praxis abilities, manual skills, graphomotor skills). In our research we consi-dered only the last scale mentioned. Raw score were converted into weighted scores and percentiles scores (a score <5° percentile was considered under the norm). The standardization sample comprised 261 children aged from 3–8 years, of which 54% were boys and 46% were girls. Internal consistency has been calculated for each of the two scales (motor schemes and adaptive cognitive functions); for both scales, α is higher than 0.75.
- Line 208 – we specified that children “appeared frustrated when the parents tried to get them away from the device”.
- We agree with the reviewer and we added this point in the limitation of the study, in the discussion section, as follow: “Finally, we did not consider other environmental factors that could impact on children digital habits, such as time spent on digital devices by their parents […]. We therefore propose to explore these aspects in future research.”
- We added “r” in the figure captions.
- We did not perform a specific comparison by age group (also because the age groups could have been arbitrary), however we correlated the age with some variables and we summarized the results in table 4.
- We agree with the reviewer and we have added the reference tables in the discussion as well.

Reviewer 2 Report
I think this study is a meaningful study that provides implications at the present time when the age of digital use is gradually getting lower.
In the method, providing information related to the reliability and validity of measurement tools would be necessary to secure the validity of the results.
It is necessary to modify the presentation method of tables so that readers can see the results at a glance. In particular, it is worth considering a method of presenting the overlapping contents (r, p, etc.) once and then presenting the data.
In the method of describing the results, overlapping references should be sorted out. For example, Table 4 and the like.
It is necessary to confirm the implications of the research results, clinical significance, and alternatives and present them in the discussion. It seems that the description is centered on the current results.
Author Response
Reviewer #2
Reviewer comment: I think this study is a meaningful study that provides implications at the present time when the age of digital use is gradually getting lower.
- In the method, providing information related to the reliability and validity of measurement tools would be necessary to secure the validity of the results.
- It is necessary to modify the presentation method of tables so that readers can see the results at a glance. In particular, it is worth considering a method of presenting the overlapping contents (r, p, etc.) once and then presenting the data.
- In the method of describing the results, overlapping references should be sorted out. For example, Table 4 and the like.
- It is necessary to confirm the implications of the research results, clinical significance, and alternatives and present them in the discussion. It seems that the description is centered on the current results.
Author response: We thank the reviewer for the comment and suggestion. We we tried to implement the manuscript as required.
- We modified the description of APCM-2, as follow: The test “Abilità Prassiche e della Coordinazione Motoria (Praxic and Motor Coor-dination Skills)—2nd Edition” (APCM-2) is an Italian test created and validated by Sab-badini [46]. This test aims to assess motor and praxis skills in children aged between 2–8 years. This tool includes 6 forms of the questionnaire, divided by age groups: 24–36 months (short version and complete versions), 37–48 months, 49–60 months, 61–72 months, and 6.1–8 years. The test includes the evaluation of two scales: (a) motor schemes (balance and coordination, sequencing, ocular movements, hand and fingers movement), and (b) adaptive cognitive functions (dynamic coordination, symbolic gestures, con-structive praxis abilities, manual skills, graphomotor skills). In our research we consi-dered only the last scale mentioned. Raw score were converted into weighted scores and percentiles scores (a score <5° percentile was considered under the norm). The standardization sample comprised 261 children aged from 3–8 years, of which 54% were boys and 46% were girls. Internal consistency has been calculated for each of the two scales (motor schemes and adaptive cognitive functions); for both scales, α is higher than 0.75.
- we agree with the reviewer and have tried to make the tables clearer at a glance, however we have not found an adequate method to guarantee the same completeness of the results. Any changes to the tables would inevitably lead to a loss of important information which could undermine the clarity of the statistical analysis performed. In order to overcome this problem, we have emphasized that significant results are expressed in bold and have created specific figures to make the main results more immediate at a glance.
- we agree with the reviewer and, to clarify the results and avoid overlapping, we have divided table 4 and created two distinct tables, one that summarizes the results of the daily time spent on digital devices (Table 4) and one that analyzes the starting age of digital device use (Table 5). We have modified the results and discussion accordingly.
- We thank and agree with the reviewer, we rearranged all the discussion section, adding the follow paragraph: “The results of our study have several practical implications, as they help to better define the possible impact of digital devices on the psychomotor development of children. Based on our results, it would be useful to monitor the time spent by children on digital tools, the age of first use, but also the modality of digital device use in order to ensure the most correct use possible. Our results highlighted also that most parents are concerned about the effect of digital tools on their children's health, but only few of them ask for the support of pediatricians, therefore it would be important to implement the communication on this topic through Information Campaigns. In the future, it would be interesting to carry out prospective studies that allow us to monitor the relationship between the use of digital devices and other development skills (academic skills, emotional and social-relational skills) during school age and adolescence, also with the aim of prevent the phenomena of digital addiction.”

Reviewer 3 Report
Thank you for the opportunity to read and review this manuscript.
This study is an interesting paper that addresses the important topic of digital devices use and fine motor skills in children between 3-6 years.
I do have some comments that the authors may wish to consider in refining the manuscript that I provide below. A few comments for consideration by authors:
1. Introduction
Many recent articles on screen time guidelines have referenced the World Health Organization's 24-hour movement guidelines instead of the American Academy of Pediatrics guidelines. Please consider making this change.
2. Materials and Methods
The details of methods of assessment (measurement) for each variable of interest are as follows:
Fine motor skills: Fine motor skills were assessed using the Movement Assessment Battery for Children-2 (MABC-2) test, which is a standardized test. Digital device use: Digital device use was assessed through a non-standardized questionnaire called the Digital Devices Questionnaire (DDQ), which was completed by parents. Provide references for the reliability and validity of the above two questionnaires (MABC-2 and DDQ).
The paper does not explicitly state how the study size was arrived at.
The exclusion criteria also included poor child compliance to perform neuropsychological evaluation and poor parental compliance. It is likely that the researchers aimed to recruit as many eligible participants as possible from each kindergarten to increase the sample size. However, without explicit information on how the study size was arrived at, it is difficult to determine whether a power analysis was conducted or whether other factors influenced the sample size. Also, The paper does not explicitly describe how missing data were addressed. It is possible that missing data were addressed by excluding participants with incomplete data from the analysis. Alternatively, missing data may have been addressed using imputation methods to estimate missing values. However, this is not explicitly stated in the paper.
(Overall, while a flow diagram was not included in this paper, its use could have improved the clarity and transparency of regarding participant progression through each study. )
3. Results
The table of findings is a confusing mix of titles and acronyms. The table needs to be reformatted to fit the paper format.
4. Discussion
Additionally, the study did not investigate all potential confounding factors that could influence the relationship between digital device use and fine motor skills. Furthermore, the study only included children aged 3-5 years, which limits the generalizability of its findings to other age groups.
No
Author Response
Reviewer #3
Reviewer comment: Thank you for the opportunity to read and review this manuscript. This study is an interesting paper that addresses the important topic of digital devices use and fine motor skills in children between 3-6 years.I do have some comments that the authors may wish to consider in refining the manuscript that I provide below. A few comments for consideration by authors:
- Introduction: Many recent articles on screen time guidelines have referenced the World Health Organization's 24-hour movement guidelines instead of the American Academy of Pediatrics guidelines. Please consider making this change.
2(a). Materials and Methods: The details of methods of assessment (measurement) for each variable of interest are as follows: Fine motor skills: Fine motor skills were assessed using the Movement Assessment Battery for Children-2 (MABC-2) test, which is a standardized test. Digital device use: Digital device use was assessed through a non-standardized questionnaire called the Digital Devices Questionnaire (DDQ), which was completed by parents. Provide references for the reliability and validity of the above two questionnaires (MABC-2 and DDQ).
2(b) The paper does not explicitly state how the study size was arrived at.
The exclusion criteria also included poor child compliance to perform neuropsychological evaluation and poor parental compliance. It is likely that the researchers aimed to recruit as many eligible participants as possible from each kindergarten to increase the sample size. However, without explicit information on how the study size was arrived at, it is difficult to determine whether a power analysis was conducted or whether other factors influenced the sample size. Also, The paper does not explicitly describe how missing data were addressed. It is possible that missing data were addressed by excluding participants with incomplete data from the analysis. Alternatively, missing data may have been addressed using imputation methods to estimate missing values. However, this is not explicitly stated in the paper. (Overall, while a flow diagram was not included in this paper, its use could have improved the clarity and transparency of regarding participant progression through each study.)
- Results: The table of findings is a confusing mix of titles and acronyms. The table needs to be reformatted to fit the paper format.
- Discussion: Additionally, the study did not investigate all potential confounding factors that could influence the relationship between digital device use and fine motor skills. Furthermore, the study only included children aged 3-5 years, which limits the generalizability of its findings to other age groups.
Author response: We thank the reviewer and agree with the comments. We have edited the manuscript as required:
- We agree with the reviewer and we have replaced the American Academy of Pediatrics guidelines with the WHO guidelines as follows: “The current recommendations from the World Health Organization's 24-hour movement guidelines claim that in children ≤ 1 year screen time is not recommended and in children between 2-4 years sedentary screen time should be no more than 1 hour (less is better) [30].” (Guidelines on Physical Activity, Sedentary Behaviour and Sleep for Children under 5 Years of Age. Geneva: World Health Organization; 2019.)
- The DDQ is a non standardized questionnaire for parents, that we have created in order to collect qualitative and quantitative data on digital device use. We have made this explicit in the methods section. We have changed the description of the APCM-2 as required as follows: The test “Abilità Prassiche e della Coordinazione Motoria (Praxic and Motor Coor-dination Skills)—2nd Edition” (APCM-2) is an Italian test created and validated by Sab-badini [46]. This test aims to assess motor and praxis skills in children aged between 2–8 years. This tool includes 6 forms of the questionnaire, divided by age groups: 24–36 months (short version and complete versions), 37–48 months, 49–60 months, 61–72 months, and 6.1–8 years. The test includes the evaluation of two scales: (a) motor schemes (balance and coordination, sequencing, ocular movements, hand and fingers movement), and (b) adaptive cognitive functions (dynamic coordination, symbolic gestures, con-structive praxis abilities, manual skills, graphomotor skills). In our research we consi-dered only the last scale mentioned. Raw score were converted into weighted scores and percentiles scores (a score <5° percentile was considered under the norm). The standardization sample comprised 261 children aged from 3–8 years, of which 54% were boys and 46% were girls. Internal consistency has been calculated for each of the two scales (motor schemes and adaptive cognitive functions); for both scales, α is higher than 0.75.
- We agree with the author, therefore we specified in the result section that “All kindergarten children were recruited (except two children with certified disabilities). All the subjects recruited agreed to participate in the study. All parents filled in the questionnaires and all children had good compliance with the APCM-2 test so there was no missing data.”
- We have modified the tables in the results as required, and replaced all abbreviations with full
- we agree with the reviewer and we added these points in the limitations of the study, as follow: “This study has several limitations, first of all the use of parental questionnaires to monitor the modalitycharacteristics of use of digital devices is not free from possible bias. Furthermore, the study only included children aged 3-5 years, which limits the ge-neralizability of its findings to other age groups. Additionally, the study did not investigate all potential confounding factors that could influence the relationship between digital device use and fine motor skills. Finally, we did not consider other environmental factors that could impact on children digital habits, such as time spent on digital devices by their parents, Furthermore, our study focuses mainly on the time of use of digital devices but not investigate the use of "educational applications" and not distin-guish between "active" or "passive" co-viewing.; We therefore propose to explore these aspects in future research.”

Reviewer 4 Report
The principal aim of this study was to explore the relationship between the fine motor skills and the use of digital tools in children between 3-5 years. Significant association was not found between the time of use of digital devices and fine motor skills, and positive correlation were found between start of use smartphone/tablet and the high fine motor skills. It is a socially important theme as a child's healthcare problem, and contains useful information.
# 1 statistical analysis
I think requires additional data analysis. Regarding the correlation between start age of smartphone/tablet use and APCM-2 score, there are other items that have significant correlations. In particular, age and start age of use DD might be closely related. Since the number of participants for which multivariate analysis is possible, I think the authors should perform multiple regression analysis and discuss the correlation.
#2 construction of the discussion section
The discussion section should be structured preferentially with sentences that corresponds to the principal aim. In this manuscript, I have the impression that there are more descriptions such as a summary and interpretation of the results of the DD questionnaire than the principal aim.
Author Response
Reviewer #4
Reviewer comment: The principal aim of this study was to explore the relationship between the fine motor skills and the use of digital tools in children between 3-5 years. Significant association was not found between the time of use of digital devices and fine motor skills, and positive correlation were found between start of use smartphone/tablet and the high fine motor skills. It is a socially important theme as a child's healthcare problem, and contains useful information.
- statistical analysis: I think requires additional data analysis. Regarding the correlation between start age of smartphone/tablet use and APCM-2 score, there are other items that have significant correlations. In particular, age and start age of use DD might be closely related. Since the number of participants for which multivariate analysis is possible, I think the authors should perform multiple regression analysis and discuss the correlation.
- construction of the discussion section. The discussion section should be structured preferentially with sentences that corresponds to the principal aim. In this manuscript, I have the impression that there are more descriptions such as a summary and interpretation of the results of the DD questionnaire than the principal aim.
Author response: We thank the reviewer for the comments and suggestions, we have edited the manuscript as follows:
- We thank the reviewer for the suggestion. We performed multiple regression analysis and we noted that the results are almost overlapping to those of the Spearman correlation analysis. However, employing the regression analysis we would lose important significant correlation, useful to give a more complete overview. The correlations between age and start age of use DD have been addressed (results section and in the tables) and the implications of this topic have been addressed in the discussion section.
- we agree with the reviewer's comment therefore we have reorganized the discussion section giving more emphasis to the relationship between fine-motor skills and the use of digital devices, that is the main result of our analysis. We reported and commented the results of the questionnaire only afterwards.

Round 2
Reviewer 4 Report
Issues I pointed out at the initial review have not been revised appropriately in this manuscript.
The start age of use of DD has positive correlation with age. (Table 5)
The start age of use of DD has positive correlation with fine motor skills. (Table 4)
The authors have described this finding is an unexpectedly found significant relationship. However, this unexpected relationship cannot be concluded to be significant unless the effect of the subject age is excluded from the multivariate analysis. If it was not significant in the multivariate analysis, it should be stated in the result section and construct discussion.
Minor issue
Page7 line 242; Table 5 and the text contradict each other. It was presented in Table 3 that father job correlates with the start age of use DD.
Author Response
Reviewer #4
Reviewer comment:
Issues I pointed out at the initial review have not been revised appropriately in this manuscript.
The start age of use of DD has positive correlation with age. (Table 5)
The start age of use of DD has positive correlation with fine motor skills. (Table 4)
The authors have described this finding is an unexpectedly found significant relationship. However, this unexpected relationship cannot be concluded to be significant unless the effect of the subject age is excluded from the multivariate analysis. If it was not significant in the multivariate analysis, it should be stated in the result section and construct discussion.
Minor issue: Page7 line 242; Table 5 and the text contradict each other. It was presented in Table 3 that father job correlates with the start age of use DD.
Author response: we thank the reviewer for the comment and suggestion. We performed the multivariate regression analysis as suggested in order to verify the effect of the age of children on the relationship “start of use of digital device – fine motor skills”. We modified the Result section and the Discussion consistently to the new result:
Results: To exclude the effect of age on this relationship we performed a multivariate regression analysis. Adding age as a covariate the relationship between start age of use and the APCM-2 weighted score is no longer statistically significant (smartphone p=0.109, ß =0.239; tablet p=0.056, ß =0.393; total p=0.297, ß =0.178)
Discussion: We found a relationship between the first age of use of digital device and fine motor skills (Table 4) (the children who used digital tools early had worse fine motor performance), however this result cannot be considered significant because affected by the effect of the age of children on this relationship.
Minor issue: We apologize for the mistake and we modified the Result section consistently (father instead of mother)